# Clinical Presentations, Cluster Analysis and Laboratory-Based Investigation of *Aspergillus* Otomycosis—A Single Center Experience

**DOI:** 10.3390/jof8030315

**Published:** 2022-03-18

**Authors:** Mila Bojanović, Aleksandra Ignjatović, Marko Stalević, Valentina Arsić-Arsenijević, Marina Ranđelović, Vladimir Gerginić, Zorica Stojanović-Radić, Ognjen Stojković, Emilija Živković-Marinkov, Suzana Otašević

**Affiliations:** 1Medical Faculty, University of Niš, 18000 Niš, Serbia; milabojanovic@yahoo.com (M.B.); drsalea@yahoo.com (A.I.); stale1995@gmail.com (M.S.); marina87nis@gmail.com (M.R.); emilijazm@gmail.com (E.Ž.-M.); 2Clinic of Otorhinolaryngology, University Clinical Center Niš, 18000 Niš, Serbia; 3Public Health Institute Niš, 18000 Niš, Serbia; 4Medical Faculty, University of Belgrade, 11000 Belgrade, Serbia; mikomedlab@yahoo.com (V.A.-A.); vladimir.gerginic@hotmail.com (V.G.); 5Department of Biology, Faculty of Science and Mathematics, University of Niš, 18000 Niš, Serbia; zorica.stojanovic-radic@pmf.edu.rs (Z.S.-R.); ognjen.stojkovic@pmf.edu.rs (O.S.)

**Keywords:** otomycosis, *Aspergillus niger*, *Aspergillus flavus*, diagnosis, biofilm production

## Abstract

Species of *Aspergillus* (*A.*) *niger* complex and *A. flavus* complex are predominant molds that are causative agents of otomycoses. The goal of this study was to investigate the clinical presentation, diagnostic procedure, and appearance of relapse in patients with *Aspergillus*-otomycosis, as well as to determine the biofilm production ability of species isolated in relapse. Thirty patients with laboratory evidenced *Aspergillus*-otomycosis followed by two check-ups (30 and 60 days after initiation of treatment with antimycotics for local application) were included in the study. For isolation and identification of *Aspergillus* spp. the standard mycological procedure was applied. Results showed very high sensitivity of microscopy, but 16.7% *Aspergillus* species required the optimal temperature of 27–28 °C for cultivation. Applied statistical cluster analysis showed a defined specific cluster/group of patients with *A. niger* complex-otomycosis. Sixty days after diagnosis and treatment initiation, six patients had a relapse, with the same species of *Aspergillus* genus being the cause. To establish the ability of biofilm production, the modified method described by Pierce and Kvasničková was performed, and all six species isolated in the relapse episode had the ability to produce biofilm. Official criteria and recommendations are needed due to the possibility of misdiagnosis, which leads to the prolongation and complication of the disease.

## 1. Introduction

Otomycosis is a superficial fungal infection of the external auditory canal (EAC). Despite the fact that it is seen in 9–27% of all the patients with EAC infection [1,2], data about microbiology protocols, prevalence, risk factors, and treatment of otomycosis is surprisingly scarce and not easily found in the literature. 

The dominant causative agents of otomycosis are fungi of *Aspergillus (A.)* and *Candida (C.)* genera; more precisely, species of *A. niger* complex, along with *C. parapsilosis* and *C. albicans* [3]. 

Infection can be acute, subacute, or chronic, and is usually unilateral. In most cases, this infection is benign, presented with aural discomfort, otorrhea, a subjective feeling of an obstruction in the ear canal, itching, dandruff of the epithelium, tinnitus, and impaired hearing. The most severe cases of otomycosis involve the perforation of the eardrum, the involvement of the middle ear or entire temporal bone infection, and are commonly associated with immunodeficiency disorders [4,5]. 

Environmental and the host factors can predispose an individual to the occurrence of this infection. However, the development of the recurrent or chronic form depends primarily on the immune status of the host and the properties of the causative agent, its antifungal susceptibility and the ability of biofilm to form [6]. The diagnosis of otomycosis is less often followed by laboratory analyses, which may include microscopy and cultivation. Cultivation is the only method that allows for the identification of fungi and is considered to be the gold standard [7,8]. Patients with a non-invasive form of infection are treated with extensive debridement of the ear canal, followed by the regular application of topical antifungal drugs. Systemic antifungal drugs, effective in *Aspergillus* infection, are kept as a last resort and used for treating forms of the disease with complications such as mastoiditis and meningitis [2].

From an epidemiological point of view, we have observed that in our region, the Nišava district (Southeastern Serbia), there are between 3000 and 3200 recorded cases of EAC infection/inflammation every year, with an annual incidence of 6–7 patients per 1000 inhabitants. However, mycological analyses are performed in only about 11.7% of them. Based on laboratory evidence, the overall prevalence of otomycosis in this region is 22.7%, and *Aspergillus* spp. were found to be the causative agent in 10.3–11.1% of all mycologically examined patients [2]. 

This high prevalence of *Aspergillus*-otomycosis, as well as the lack of official guides for diagnostic procedure and treatment protocols for this infection, compelled us to explore laboratory diagnostic procedures, using the cluster analysis, to determine the most common phenotype of this infection and monitor the effectiveness of the treatment. Additionally, in relapsed patients with *Aspergillus* otomycoses, causative species were analyzed for their ability to produce biofilm.

## 2. Materials and Methods

### 2.1. Patients 

In this retrospective pilot study, we used patients’ data from the database of the laboratory for mycology at the Public Health Institute of Niš, Serbia. The investigation was conducted in the period from the beginning of 2020 until the end of 2021. Inclusion criteria for the defined study group were:

(i) Patients > 18 years old with *Aspergillus*-otomycosis who had a microbiological diagnostic procedure involving the bacteriological and mycological analyses of EAC material; (ii) patients with *Aspergillus* otomycosis, which was defined as two consecutive positive results of the same *Aspergillus* species, and a negative test for pathogenic and conditionally pathogenic bacteria in EAC samples; (iii) patients who were being treated with antimycotics for a local application; (iv) patients who had two control mycological examinations; the first one after 30 days and the second one 60 days after the treatment initialization; and, (v) patients who completed questionnaires that included general data, the questions of present symptoms and signs of infection, as well as possible risk factors.

### 2.2. Microbiological Analyses

All ear samples were acquired with four cotton swabs. One was prepared with standard saline solution for microscopic examination (wet-mount preparations for direct microscopic detection of fungal elements such as yeast and hyphal forms). Two samples were inoculated on two Sabouraud Dextrose Agar (SDA) (Liofilchem Diagnostic, Roseto degli Abruzzi, Italy), incubated at 37 °C for up to seven days at 26–28 °C for seven to days. Cultures were checked on alternate days. *Aspergillus* spp. were identified based on macroscopic and microscopic morphological characteristics. Genus and species were identified based on macroscopic (features, appearance and color of colonies) and microscopic—*Aspergillus* sp. type of sporulation characteristics. *A. niger* complex macroscopic characteristics: colonies first white, then dark brown to black, reverse light brown; microscopic characteristics: conidial heads are biseriate, large, dark brown, radial, with flask-shaped phialides on metulae with subspherical phialoconidia; *A. flavus* complex macroscopic characteristics—colonies first yellow but quickly becoming bright to dark yellow/green; microscopic characteristics—conidial heads with uniseriate or biseriate arrangement of phialides with hyaline globose to subglobose phialoconidiae [9]. A fourth sample of material was used for the bacteriological analysis performed by standard bacteriological analyses [inoculation of material on blood agar and MacConkey agar (Oxoid Ltd., Basingstoke, UK), incubated on 37 °C for 24 h)].

### 2.3. Examination of Ability in Biofilm Production

Examination of biofilm formation by *A. niger* complex and *A. flavus* complex was performed according to the method described by Pierce et al. [10] and Kvasničková et al. [11] with slight modifications. The experiment was performed under static conditions in 96-well microtiter plates. The initial spore suspension was used for the inoculations of wells, containing 200 µL of RPMI 1640 supplemented with 0.8% (*w*/*v*) glucose, in aliquots adjusted to achieve the final spore concentration of ~10^5^ spores/mL. The microtiter plates were covered with lids and incubated at 37 °C for 72 h. After the incubation period, each well was washed three times with PBS, dried and filled with 200 µL of 0.1% CV (crystal violet) solution. The plates were incubated for 20 min at room temperature. Thereafter, plates were washed with physiological saline (0.85% NaCl), filled with 200 µL of 96% (*v*/*v*) ethanol and incubated for 45 min at room temperature. The obtained solutions were transferred into new microtiter plates, and the absorbance of the solutions was measured at 595 nm using an ELISA reader (Multiscan Ascent, Labsystems, Vantaa, Finland). According to the ability to produce biofilm, the tested strains were divided into four groups: none, weak, moderate and strong biofilm producers [12].

### 2.4. Statistical Analysis 

Descriptive statistics were used to report the presence of signs and symptoms and risk factors (count and percentage). Continuous data are presented as arithmetic mean and standard deviation (SD). Agglomerative hierarchical cluster analysis was used to study the similarity of signs, symptoms, and risk factors with the Euclidean distance used to measure the similarity between variables (Ward’s method). In cluster analysis the following demographic, clinical and laboratory evidence data were used: gender, older than 60 years, severe pain, feeling of EAC fullness/obstruction, tinnitus, black discharge, impaired hearing, isolated species- *A. niger* complex, and *A. flavus* complex. A clustering analysis was graphically reported with the dendrogram. The patients were divided into two clusters based on cluster analysis. The frequency of demographic and clinical variables was compared among clusters using a chi-squared test and a Fisher’s test. All statistical analyses were performed using R software (version 3.4.3; The R Foundation for Statistical Computing, Austria). 

## 3. Results

The study involved 30 patients with *Aspergillus*-otomycosis (16 males and 14 females); the mean age of the study group was 59.27 years (SD 18.75 years) (Min 21, Max 88 years), and the mean duration of infection was 40.83 days (SD 27.04; Min 10; Max 150). In this defined group, causative agents were species of *A. niger* complex (66.7%) and *A. flavus* complex (33.3%). Only one patient had an infection of both EAC, caused by species of *A. flavus* complex. 

Based on information obtained from the questionnaire, the most common risk factors are the usage of antibacterial ear drops (63.3%), chemicals for hygiene, such as soaps, shampoos, boric acid, povidone-iodine, hydrogen peroxide and other antiseptics (50.0%), followed by the frequent use of headphones (27.6%). Other predisposing factors are presented in Figure 1.

Regarding applied diagnostic procedures, microscopy showed very high sensitivity (90.0%). As for cultivation, in 83.3% of the incubated samples, molds grew at both temperatures, but five species (*A. niger* complex) required a temperature of 27–28 °C for cultivation. 

In the initial examination, almost all of our patients reported symptoms of high intensity—severe pain (100%), itching (96.7%), a subjective feeling of obstruction of the ear canal (86.7%), otorrhea (83.3%), tinnitus (83.3%) and impaired hearing (70.0%). To establish the phenotype of this infection we used cluster analysis (Figure 2). This agglomerative hierarchical cluster analysis divided patients into two clusters. Cluster 1, consisting of 20 patients, was characterized by the following parameters: 100.0% *A. niger*, 100% severe pain, 95.0% subjective feeling of obstruction of the ear canal, 95.0% tinnitus, 75.0% black discharge, 60.0% men, 60% frequent ear cleaning, 45.0% use of antiseptics. Cluster 2, consisting of 10 patients, was characterized by the following parameters: 100.0% *A. flavus*, 100% severe pain, 100% subjective feeling of obstruction of the ear canal, 80% tinnitus, 60.0% impaired hearing, 60.0% women, 70% frequent ear cleaning, 70.0% use antiseptics and 40.0% comorbidities. The comparison of these two clusters showed that the mean duration of infection was longer in Cluster 2 than in Cluster 1, but without statistical significance (52.00 ± 38.82 vs. 35.25 ± 17.43, *p* = 0.322). However, a black secretion was statistically significantly more common in Cluster 1 (75.0% vs. 0.0%, *p* < 0.001), but for Cluster 2, comorbidities were statistically significantly more frequent (40.0% vs. 0.0%, *p* = 0.008).

After treatment with topical antifungal drugs [Clotrimazole (CLO) ear drops—eight patients, Nystatin (Ny) ointment—eight patients; Naftifine cream (NAF)—14 patients, once per day, in the evening, for 14 days (with a recommendation to continue the same treatment for the next two weeks if symptoms and signs do not subside)], in the second laboratory examination/the first control, 30 days after treatment initiation, 33.3% patients had low to moderate intensity symptoms (mostly subjective feeling of obstruction of the ear canal and itching). All of the patients reported that the pain was reduced after only two applications of antifungals, and 10 patients with persisting symptoms were on the following treatment: CLO, six patients, Ny, two patients, NAF, two patients. Mycological analyses (microscopy and culture) showed negative results in all patients. 

However, in the third laboratory examination/the second control (60 days after treatment initiation), six patients (4 CLO, 1 Ny, 1 NAF) had positive findings of *Aspergillus* spp., the same species as in the first examination, with the presence of moderate-intensity symptoms (a subjective feeling of obstruction of the ear canal and itching). Two patients had symptoms of the same intensity without laboratory evidence of otomycosis. Another 22 patients reported low intensity symptoms (5/30) or no symptoms at all (17/30). All *Aspergillus* species which caused a relapse (four species of *A. niger* complex and two species of *A. flavus* complex) were examined for biofilm formation, and all of these isolates had biofilm formation ability. More precisely, four isolates of *A. niger* complex and one isolate of *A. flavus* had the ability of strong biofilm production, while one isolate of *A. flavus* had the ability of weak biofilm production. 

## 4. Discussion

The most common problem in distinguishing bacterial from fungal infections is reflected in the similarity of symptoms and clinical findings without specifics that would allow differentiation during a clinical examination [13]. In order to overcome this problem, we tried to determine the phenotype of the disease by cluster analysis in order to demonstrate specific indicative symptoms and clinical findings in patients with *Aspergillus*-otomycosis. Regarding the applied statistical method, in the case of *A. niger*-otomycosis, we can highlight that black discharge with all EAC infection symptoms may be a specific indicator of otomycosis due to *A. niger* infection. However, cluster analysis established one more cluster that included other patients with *A. flavus* and we do not have specifics that can predict infections caused by molds. Although some *A. flavus* strains are found to be able to produce melanin [14], possibly leading to darker colored colonies, in this research we did not encounter one. Attempts to determine the phenotype of a particular disease by statistical methods, like in our previously conducted examination, showed that, in a certain percentage of patients, it is impossible to determine the cause without laboratory examination [15].

In this study, microscopy showed very high sensitivity (90%), implying that wet-mount preparations are cost-effective, easy to use, and the fastest way to prove fungal elements (conidia and hyphal forms) in patient’s material. The higher sensitivity of microscopy recorded in our study, compared to the results of other authors, can be explained by the fact that they referred to the sensitivity of this method in otomycoses caused by yeast and molds, but here we focused only on *Aspergillus*-otomycoses [7]. As for the cultivation process, we can highlight that 83.3% of molds grew at both incubation temperatures, and five samples (*A. niger* complex) required a temperature of 27–28 °C. This finding indicates the possibility of false results if the procedure does not include incubation at the temperature prescribed for molds and warrants implementation of appropriate culture protocols to enhance fungal detection [16,17].

Well-known risk factors, such as excessive use of topical antibiotics or antiseptics as well as exaggerated use of soaps and shampoos while maintaining the hygiene of EAC were identified in a high percentage of our patients. Furthermore, we noted that frequent ORL examinations and rinsing of EAC (16.7%), predisposing diseases (chronic ORL infections, diabetes mellitus, systemic lupus erythematosus)—13.3%, and corticosteroid ear drops usage (13.3%) were other prominent risk factors. A significant portion of our patients often used headphones (27.6%), and 90% of examinees used mobile phones regularly, both of which could be treated as risk factors, considering our past findings that 26% of mobile phones were contaminated by fungi, mainly by *Aspergillus* spp. molds [18]. 

A standard treatment regimen for otomycosis has not yet been established. Some studies focused on the efficacy of different antifungal drugs; however, most of them followed the course of the treatment without laboratory-based evidence. Multiple in vitro studies have examined the efficacy of various antifungal agents, yet there is no consensus regarding the most effective one that is suitable for treating both yeast and molds equally [19,20]. Several different antimycotics (nystatin, clotrimazole, miconazole, fluconazole, tolnaftate, naftifine, bifonazole, econazole, etc.) have been reported to be effective and can be applied locally as part of the otomycosis treatment. However, clinicians usually prescribe antimycotics empirically based on their experience and availability of officially approved antifungal drugs. Consequently, our patients were on treatment with CLO, Ny and NAF, as they are antifungal drugs readily available in our region. The recommended treatment duration was up to 14 days, which should be continued for another two weeks if the symptoms and signs are not remedied. 

Antifungal treatment follow-up showed that most of the patients with recorded relapse of the infection (six out of eight) were treated with CLO, where four of them had laboratory evidence of recurrent otomycosis. That result is not surprising, since CLO is an antifungal agent most widely used, yet there is no conclusive laboratory evidence of its efficacy against the *Aspergillus* species [8,21]. Nystatin was the other applied antifungal, which is recommended and has been approved in some countries for the topical treatment of otomycosis. Regardless of previous reports of nystatin’s low efficacy in some cases of *Aspergillus* otomycosis [4,22], our study showed a good result against *Aspergillus* mold, with only one recorded relapse. The third prescribed antimycotic was naftifine, an allylamine derivative for topical administration which, besides its antifungal effect, may have certain anti-inflammatory properties [23]. This drug seemed to be a good option for treatment, as we found only one patient with relapse. In total, during 60 days of monitoring, we recorded recurrence of the infection in 20% of our patients, which was confirmed in the laboratory. Besides treatment monitoring and consideration of the used antimycotics, the ability of biofilm production was examined in all of the species that caused relapse of the disease. Five of these species showed the ability of strong biofilm production, which could be another reason for treatment inefficiency, since microorganisms are protected from the action of antifungals in the biofilm [24]. 

The main limitation of this study is the small number of involved patients. This disadvantage is a consequence of the COVID-19 pandemic and general education and extensive campaigns which warned people to avoid going to health facilities unless necessary. Consequently, a number of patients gave up visiting the doctors. However, based on our results, we could conclude that dominant risk factors are antibacterial drugs and/or antiseptic overuse; cluster statistical analysis demonstrated a specific phenotype of the disease where clinical symptoms and signs of EAC infection, followed by black discharge, is very indicative for *Aspergillus niger* complex otomycosis, and this may be helpful for clinicians. As for laboratory diagnostic procedures, it can be highlighted that microscopy had high sensitivity, but cultivation is also a necessary procedure for molds. Furthermore, after treatment some patients had a relapse caused by the biofilm producing molds, which can be a reason for treatment ineffectiveness. Additionally, our pilot study revealed many problems regarding otomycoses that have to be solved in future investigations. Most importantly, we need official guides in diagnosis and treatment procedures which define the necessity for laboratory examination and in vitro antifungal susceptibility testing. Moreover, laboratory capacity for mycological analyses must be appropriate for isolation and identification of molds because that is the best way to avoid misinterpretation of the results [25]. Defining and applying adequate molecular techniques in the identification of fungi at the species level by DNA sequencing is also needed in the future.

## Figures and Tables

**Figure 1 jof-08-00315-f001:**
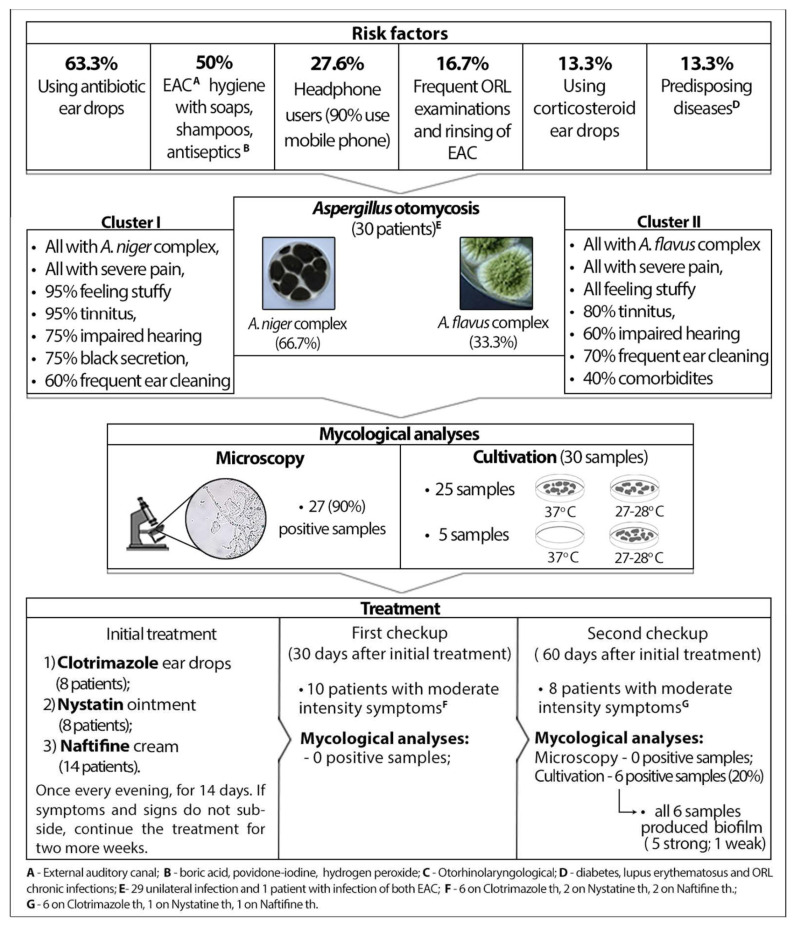
Summary of the results of this study.

**Figure 2 jof-08-00315-f002:**
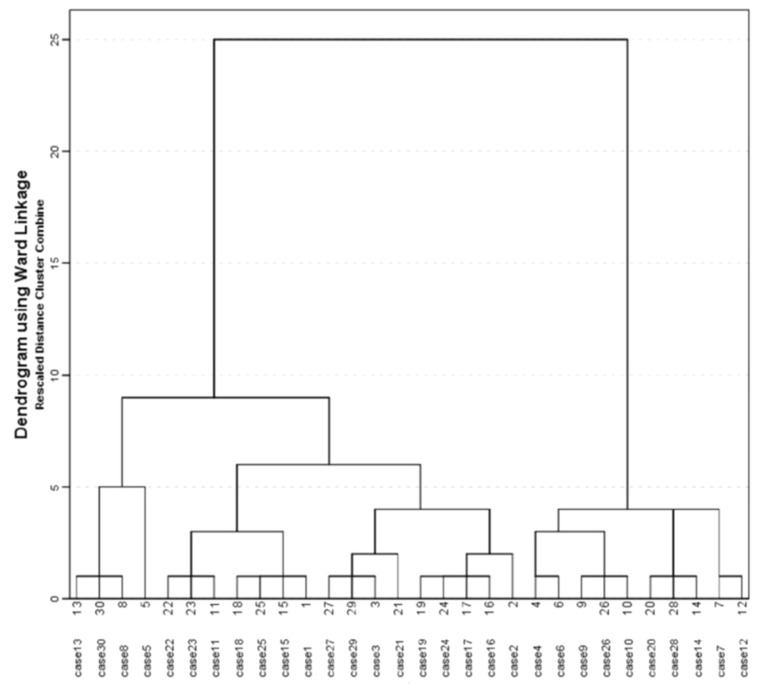
Dendrogram of the patients with *Aspergillus*—otomycosis.

## Data Availability

The data presented in the study are available on request from the corresponding author.

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
