# Peer review of "Clinical Presentations, Cluster Analysis and Laboratory-Based Investigation of *Aspergillus* Otomycosis—A Single Center Experience"

_jof, 2022, doi:10.3390/jof8030315_

Round 1
Reviewer 1 Report
Dear Authors,
I fully agree, that "data on microbiology protocols, prevalence, risk factors, and treatment of otomycosis is surprisingly scarce and not easily found in literature". I am convinced that the described data may be interesting and useful to many researchers and doctors. The graphic abstract is very well structured and legible. My objections are raised by biofilm research - we learn from the title that the study concerns "Laboratory Based Investigation of Biofilm Forming Aspergillus caused Otomycosis" - nevertheless, the ability to produce biofilm was examined only for a few strains (isolates from recurrent infections). The title should be changed or a larger number of strains should be included in biofilm examinations. The lack of comparison between biofilms in two groups of isolates under study, make the interpretation of the results difficult. On the other hand, extending the study by the identification of fungi to the species level (DNA sequences) and/or assess their drug susceptibility could be more valuable than the biofilm studies presented.
Author Response
Dear reviewer, thank you for your consideration that the described data may be interesting and valuable to many researchers and doctors, and thank you for all of your benevolent suggestions that were accepted for the revised version of the manuscript.
All of the changes are marked in blue color.
Following your suggestion:
- we changed our previously title Laboratory Based Investigation of Biofilm Forming Aspergillus caused Otomycosis: Clinical Presentation and Cluster Analysis
to: Clinical presentations, cluster analysis and laboratory based investigation of Aspergillus- otomycosis - A single-center experience as it is more appropriate to the topic of paper
- in Discussion part, we point out the importance of future research in the field of molecular identification of Aspergillus spp., causative agents of otomycosis to the species level, as well as the significance of in vitro drug susceptibility testing.
Thank you for your revision.
Sincerely,
Suzana Otašević.
Reviewer 2 Report
The study aims to investigate the clinical presentation and improvements in the diagnostic procedures for patients with Aspergillus otomycosis, as well as ability of biofilm production of the fungal species isolated in relapse.
Thirty patients with evidenced Aspergillus-otomycosis followed by two check-ups were included in the study. The research was approved by the Ethical Committee of the Faculty of Medicine and the Public Health Institute at the University of Niš in 2014. The period and the amplitude of time for the study should be mentioned. This number is not very high, and we are in 2022.
The following points would be addressed before definitive acceptance
- Lines 98-102: Some more details about the microscopic morphological characteristics of Aspergillus or the standard bacteriological analyses would be provided or referenced.
- Concerning the use of antifungal for otomycosis, the manuscript describes some confusing information. Line 59: It is presented that systemic antifungal drugs (itraconazole, voriconazole) are effective in However, at lines 167-168 and figure 1: Patients were treated with the following topical antifungal drugs [Clotrimazole (CLO) ear drops, 8 patients; Nystatin (Ny) ointment, 8 patients; Naftifine cream (NAF) 14 patients. Those antifungals are different of the agents mentioned before. Moreover, at line 231, it is discussed that several different antimycotics (clotrimazole, miconazole, fluconazole, tolnaftate, naftifine, bifonazole, econazole) have been reported effective; finally, at line 238, it is stated that clotrimazole is the antifungal agent most widely used despite no evidence of its efficacy against Aspergillus species.
Even understanding the reasons of possible differences between antifungals used in previous studies and in this study, some extra discussion about the choice and use of antifungals would be informative and helpful. For instance, the criteria for using Clotrimazole, Nystatin or Naftifine, and some previous experience about these agents in comparison to itraconazole and voriconazole, included at the introduction but never used. I think that the main final conclusion is right and helpful. Results indicate the necessity of official guides in diagnosis and treatment procedures, as well as of laboratory examination to avoid misinterpretation of results and more efficient treatment. It is sure that laboratory capacity for mycological analyses must be appropriate for isolation and identification of fungi.
- A suggestion concerning Figure 1 and the mention to black secretion. Is that because the ability of niger of synthesizing dark melanin pigments? Please, if possible, give some details, as A. flavus seems to be unable to synthesize melanin pigments.
- Line 140: About common risk of factors. The use of headphones is a reasonable risk factor, but the use of povidone iodine, H2O2 or other antiseptics is not so reasonable. Clarify please the possible reason for those antibacterial or antifungal agents. They are also used in oral hygiene, and they should not be a risky factor for fungal infection.
Author Response
Dear reviewer, thank you for your helpful suggestions which we took into account while revising the paper. Thank you for emphasizing the importance of this fungal infection and problems which has to be solved in future research. All of the changes are marked in yellow.
Following your instructions and requirements:
- we added the period when we conducted our investigation.
- in the revised version of the Material and methods chapter , we described macroscopic and microscopic morphological characteristics of Aspergillus that were used to determine isolated species.
- additionally, we added our protocol for bacteriological analysis.
- most importantly, we tried to be clearer and more precise in the discussion regarding the choice of the treatment of our patients. We hope that we have succeeded in highlighting the problem - lack of official protocols and guidelines and, consequently, doctors' empirical choice of antifungals for the treatment in the group of the patients included in our study.
- Finally, we point out that overuse of antibiotics and antiseptics could be risk factors.
Thank you for your revision.
Sincerely,
Suzana Otašević.
Round 2
Reviewer 1 Report
The text is ok, except of a few spelling errors e.g. clotrimasole/clotrimazole), references.
Author Response
Thank you for your suggestions. We have made requested changes ( "clotrimasole" to "clotrimazole").